# Research on Nonlinear Behavior of Local High-Performance Concrete Beam–Column Connections

**DOI:** 10.3390/ma17010107

**Published:** 2023-12-25

**Authors:** Zhiqiang Xu, Jianbing Yu, Yufeng Xia, Chaojun Jiang

**Affiliations:** College of Civil Science and Engineering, Yangzhou University, Yangzhou 225127, China; xzq199812161216@163.com (Z.X.); yfxia42@163.com (Y.X.); jiangchaojun0113@163.com (C.J.)

**Keywords:** seismic performance, finite element analysis, beam–column joints, hysteretic model, parameter analysis

## Abstract

The seismic performance index of prefabricated structures is generally obtained via experimental analysis. However, in experimental research, it is impossible that every influencing factor can be taken into account. Therefore, the finite element analysis method can be used as a supplementary method for experimental research to carry out parametric analysis of joints. Based on this test, a hysteretic model of steel bars is developed on the ABAQUS platform; meanwhile, the model is used to simulate the seismic analysis of the proposed local reinforced joints. The hysteresis curve obtained via simulation exhibits a high degree of coincidence with the experimental results. Based on the validated model, a detailed parameter analysis of prefabricated local reinforced concrete frame joints is carried out. The analysis results illustrate that the axial pressure ratio at the top of the column has a minimal impact on the joint’s performance. Decreasing the stirrup ratio within the core region, enlarging the diameter of the PC steel bar, and increasing the concrete strength that is poured in the keyway and the core region can raise the cumulative energy consumption of the joints, thereby reducing the damage degree of other units and improving the maximum bearing capability of the joints.

## 1. Introduction

A prefabricated assembly structure is favored by more and more countries and enterprises because of its high construction efficiency, low pollution emission, and recyclability [1,2,3]. The prefabricated concrete frame system is one of the most popular structural forms; the precast concrete frame structure has also developed rapidly in recent years. However, in China, it can be applied mainly to the post-pouring integral prefabricated concrete frame structure. The so-called post-pouring integral type refers to the prefabrication of beams and columns in the component factory; then, the precast columns and precast beams are spliced at the construction site. The beam–column connection not only bears the load of the column top but also redistributes the bending moment transmitted from the beam end. Therefore, it is the most vulnerable part of the prefabricated concrete frame structure [4,5]. Therefore, the seismic behavior of beam–column connections plays a crucial role in determining the reliability of precast frame structures [6,7]. To improve the shear resistance of the structure, most designers generally increase reinforcement in the joint area [8], resulting in the steel bar being very dense, thereby reducing the bonding force of concrete. Haidei [9], Parastesh [10], Lee [11], and Eom et al. [12] conducted experimental research to study the difference in the mechanical properties between prefabricated concrete joints and cast-in-place joints under earthquake action. Their research results showed that reasonably designed prefabricated joints can reflect a seismic performance superior to the cast-in-place overall structure and can meet the seismic requirements of high-intensity earthquake areas. To achieve the recycling of prefabricated structures after earthquakes, Morgen [13], Baran [14], and Han [15] installed energy dissipation devices on prefabricated structures and conducted low-cycle loading experiments. The test showed that the prefabricated structure with an energy dissipation device showed an excellent self-recovery ability and energy dissipation capacity, which can meet post-earthquake repair and recycling demands.

With the wide application of high-performance materials in the construction field, some scholars have found that pouring high-performance concrete in the connection area of assembled structures can improve the reliability of beam-to-column connections. Vasconez [16], Andre [17,18], Zheng [19], and Dinesh et al. [20,21,22,23] proposed some approaches to cast fiber-reinforced concrete into the post-pouring zone of a precast structure and carried out some corresponding experiments. It was found that fiber-reinforced concrete can enhance the synergy between steel and concrete and improve the bending resistance and failure resistance of joints. Choi et al. [24,25] and Yuan et al. [26,27,28] applied an engineered cementitious composite (ECC) to the connection area of a prefabricated structure and carried out many cyclic loading experiments. The test results showed that pouring ECC can reduce the stirrup ratio and enhance the shear failure resistance and energy consumption capacity of the structures. 

The finite element analysis method has become another important research method in studying structural properties besides the typical methods of experimental researches. For example, when studying the seismic performance of prefabricated frame structures, it is difficult to take into account every influencing factor. Therefore, the finite element model can be established using the results obtained from the test to further analyze the joints. Yu et al. [29,30] and Sarma et al. [31,32,33,34] simulated and analyzed the seismic capacity of prefabricated reinforced concrete frames in earthquakes with finite element software. The numerical method showed a high calculation accuracy. Therefore, the advantages of the finite element simulation software can be used to investigate the capacity of precast constructures to resist seismic disasters under different parameters at a lower cost.

For the purpose of enhancing the seismic performance of beam–column joints, a kind of prefabricated local reinforced concrete joint is proposed in this paper. Through the combination of experimental and numerical analyses, the mechanical properties of prefabricated joints under different parametric conditions were studied. The overall strength of the structure is improved by pouring the panel area and the keyway with reactive powder concrete (RPC), which has superior tenacity, low porosity, and ductile properties. When the precast beam was made, in order to stimulate better synergy between different concretes, the shear key was set inside the precast beam keyway. PC steel bars (steel bars for prestressed concrete) with higher strength and flexibility were used as the stressed steel bars in the bottom position of the prefabricated beams. Good flexibility can solve the problem of steel bar collision in the core area when the joints are assembled. As the plastic deformation capacity of PC steel bars is not as good as that of the ordinary steel bars, L-shaped steel was adopted to improve the structural energy consumption and shock absorption capacities, as shown in Figure 1.

## 2. Experimental Design

### 2.1. Size Design of Test Specimen

The joints were designed based on the principle of “strong joints and weak members, strong shear and weak bending”. The dimensions of the prefabricated column were set as follows: the cross-section dimension of the prefabricated column was 550 mm × 550 mm, the height of the prefabricated upper column was 1.30 m, the height of precast lower column was 1.0 m. The longitudinal steel bars in the precast column were 12B20, and the stirrups in the encryption area of the prefabricated column were B8@100, the rest of the stirrups were B8@200. The cross-section dimension of the precast beam was 300 mm × 550 mm, the top space of 120 mm was the laminated layer. The precast beams on either side of the precast column were 2.0 m in length, and the end of the precast beam had a keyway with a length of 500 mm, which was convenient for the assembly of the components. The bottom of the precast beam was reinforced by three D12.6PC, which extended 500 mm beyond the beam end and were bent 250 mm vertically upwards. The stirrups in the keyway range were B8@100, the remaining stirrups in the precast beam were B8@200. Three B20 longitudinal bars were used in the upper part of the beam. During the assembly, an L-shaped angle steel was set by bolts to enhance joints’ energy consumption ability, the diameter of the bolt was 10 mm and the length was 200 mm. The composite layer of the precast beam was poured with ordinary concrete, and casted the remaining post-pouring area with RPC, detailed information can be seen in Figure 2.

### 2.2. Test Loading Scheme

The horizontal reciprocating load was applied to both sides of the prefabricated column top by controlling the hydraulic servo control system (MTS). The axial pressure ratio was set as 0.4. The loading situation can be seen in Figure 3.

The loading system can be seen in Figure 4. The loading system adopted the drift ratio control system recommended by the American Concrete Institute [35]. Before the start of the formal test, a preload of 1–2 mm displacement was carried out. In the formal test, the drift ratios were controlled to be 0.2%, 0.25%, 0.35%, 0.5%, 0.75%, 1.0%, 1.5%, 2.0%, 2.75%, 3.5%, and 4.25%, and each stage was repeated for three times. When the load-resistance of the component reached its peak value and decreased to 80% of the ultimate strength, the load test was stopped. The displacement control loading process is shown in Table 1.

### 2.3. Hysteresis Characteristics

The specimen is a nonlinear composite material composed of steel bars and concrete. The elastic deformation capacity is very small, and there is a lag phenomenon. After repeated loading, the load–displacement curve forms a ring, the hysteresis curve [36]. The hysteresis curve is a comprehensive curve reflecting the integral behavior of the structure under repeated loads. The hysteresis curve of the precast joint specimen in this test adopted the load–displacement curve, as shown in Figure 5.

Figure 5 shows that at the beginning of the test loading, the hysteresis curve changed linearly, and the structure had a strong self-centering ability. Once the structure entered the elastic-plastic stage, the beam surface began to crack, and residual deformation increased. This resulted in a stronger ability to consume energy. When the displacement came to 42.4 mm, concrete spalling occurred at the junction of precast beam end and precast column. Due to the lower plastic deformation ability of PC steel bars compared with ordinary steel bars, there was no distinct yield stage. When the concrete at the beam end peeled off, the L-shaped steel still ensured the structure having a strong bearing capacity. From the start of the loading process until the displacement value reached 55.1 mm, it can be seen that the curves of the three cycles at all levels were highly coincident, it showed that L-shaped steel enhanced the self-centering ability of the component and can maintain good performance during the cyclic loading process. With the displacement reaching 73.24 mm, the beam–column connection reached the maximum bearing capacity, the hysteresis loop appeared stratification, and the self-centering ability of components was gradually weakened. At the level of 99.8 mm, the maximum bearing value of the component decreased for the first time and the hysteresis loop stratification became more obvious. The concrete at junction of beam end and column was seriously broken under compression, and the crack width became the largest. The pinching of the hysteresis curve became more and more obvious, an inverted s-shaped was formed at last. When the displacement value reached 121.1 mm, the concrete near the L-shaped angle steel was crushed, and the recovery capacity of the component was further reduced. After a load cycle, the component was damaged and the test was terminated. On the whole, precast component had full hysteretic curve and good seismic resistance.

## 3. Establishment and Analysis of the Finite Element Model

### 3.1. The Material Constitutive Model

#### 3.1.1. The Ordinary Concrete Constitutive Model

The constitutive model of the ordinary concrete material in this paper adopted the model recommended by the Code for Design of Concrete Structures [37] (GB50010-2010). The specific formulas are as follows:

Uniaxial tensile constitutive of concrete:(1)$$\sigma =(1-d_{t})E_{c}\varepsilon$$
(2)$$d_{t}=\left\{\begin{array}{c} 1-p_{t}(1.2-0.2x^{5})\\ 1-\frac{p_{t}}{\alpha_{t}{\left(x-1\right)}^{1.7}+x} \end{array}\right.\begin{array}{c} ,x\leq 1\\ ,x>1 \end{array}$$
(3)$$x=\frac{\varepsilon }{\varepsilon_{t,r}}$$
(4)$$p_{t}=\frac{f_{t,r}}{E_{c}\varepsilon_{t,r}}$$

Uniaxial compression constitutive of concrete:(5)$$\sigma =(1-d_{c})E_{c}\varepsilon$$
(6)$$d_{c}=\left\{\begin{array}{c} 1-\frac{p_{c}n}{n-1+x^{n}}\\ 1-\frac{p_{c}}{\alpha_{c}{\left(x-1\right)}^{2}+x} \end{array}\right.\begin{array}{c} ,x\leq 1\\ ,x>1 \end{array}$$
(7)$$p_{c}=\frac{f_{c,r}}{E_{c}\varepsilon_{c,r}}$$
(8)$$n=\frac{E_{c}\varepsilon_{c,r}}{E_{c}\varepsilon_{c,r}-f_{c,r}}$$
(9)$$x=\frac{\varepsilon }{\varepsilon_{c,r}}$$

#### 3.1.2. The RPC Constitutive Model

The specimens used in this paper were cast with RPC in the core region and keyway. The RPC constitutive model proposed by Bu et al. [38,39,40] was adopted herein.

Uniaxial compression constitutive of RPC:(10)$$y=\left\{\begin{array}{c} Ax+(5-4A)x^{4}+(3A-4)x^{5}\\ \frac{x}{a(x-1)} \end{array}\right.\begin{array}{c} ,0\leq x\leq 1\\ ,x>1 \end{array}$$

Uniaxial tensile constitutive of RPC:(11)$$y=\left\{\begin{array}{c} Ax+(3-2A)x^{2}+(A-2)x^{3}\\ \frac{x}{a{\left(x-1\right)}^{1.7}x} \end{array}\right.\begin{array}{c} ,0\leq x\leq 1\\ ,x>1 \end{array}$$

#### 3.1.3. The Concrete Damage Plasticity Model

The damage plasticity model provided by ABAQUS was utilized to simulate the performance of concrete materials. Directly applying the damage evolution parameters provided in the Code for Design of Concrete Structures [37] (GB50010-2010) to the damage plasticity model in ABAQUS CAE 2021 software usually leads to the wrong calculation result or a convergency problem. After many verifications, it is found that the damage factor algorithm based on the energy method proposed by Sidoroff [41] is more suitable for the damage plasticity model in ABAQUS. The mathematical expressions are listed as follows:(12)$$d=1-\sqrt{\frac{\sigma }{E_{0}\varepsilon }}$$

The inelastic strain in the damage plasticity model should ensure that the corresponding ${\tilde{\varepsilon }}_{c}^{\mathrm{pl}}$ and ${\tilde{\varepsilon }}_{t}^{\mathrm{pl}}$ are all greater than 0 and exhibit an increasing trend with the rise in damage factor, the compression expression is as follows:(13)$$\left\{\begin{array}{l} {\tilde{\varepsilon }}_{c}^{in}=\varepsilon_{c}-\varepsilon_{oc}^{el}\\ \varepsilon_{oc}^{el}=\frac{\sigma_{e}}{E_{0}}\\ {\tilde{\varepsilon }}_{c}^{pl}={\tilde{\varepsilon }}_{c}^{in}-\frac{d_{c}}{\left(1-d_{c}\right)}\frac{\sigma_{c}}{E_{0}} \end{array}\right.$$

The mathematical expression of tensile cracking strain is as follows:(14)$$\left\{\begin{array}{l} {\tilde{\varepsilon }}_{t}^{ck}=\varepsilon_{t}-\varepsilon_{ot}^{el}\\ \varepsilon_{ot}^{el}=\frac{\sigma_{t}}{E_{0}}\\ {\tilde{\varepsilon }}_{t}^{pl}={\tilde{\varepsilon }}_{t}^{ck}-\frac{d_{t}}{\left(1-d_{t}\right)}\frac{\sigma_{t}}{E_{0}} \end{array}\right.$$

When the load changes from tensile stress to compressive stress under a repeated loading condition, the elastic stiffness of concrete can be partially restored to some extent. Lubliner et al. [42,43] introduced the concept of stiffness recovery to improve it by adding a damage factor to control the stiffness recovery of compression and tension under a reverse load, the stiffness recovery factor $w_{t}$ and $w_{c}$ are introduced to control the recovery of stiffness. It is assumed that:(15)$$E=(1-d)E_{0}$$
(16)$$\left(1-d)=(1-s_{t}d_{c})E_{0}(1-s_{c}d_{t}\right)$$
(17)$$\left\{\begin{array}{c} s_{t}=1-w_{t}r^{*}(\sigma_{11})\\ s_{c}=1-w_{c}(1-r^{*}(\sigma_{11})) \end{array}\right.\begin{array}{c} ,\;0\leq w_{t}\leq 1\\ ,0\leq w_{c}\leq 1 \end{array}$$
(18)$$r^{*}=H\left(\sigma_{11}\right)=\left\{\begin{array}{c} 1\\ 0 \end{array}\right.\begin{array}{c} ,\sigma_{11}>0\\ ,\sigma_{11}<0 \end{array}$$

$\omega_{t}\;$= 1 or $\omega_{c}\;$= 1 indicates that the concrete can completely restore the elastic modulus at the last unloading, $\omega_{t}\;$= 0 or $\omega_{c}\;$= 0 indicates that the concrete cannot restore the elastic modulus at the last unloading, as shown in Figure 6.

#### 3.1.4. The Steel Constitutive Model

The ABAQUS finite element software does not consider the synergistic mechanism between steel bars and concrete in the nonlinear analysis. As a result, the simulation results may not accurately reflect this experimental phenomena. Fang [44] developed a subroutine that can reflect the change in the unloading stiffness of steel bars. When using the software of ABAQUS, the synergistic effect between steel and concrete is considered by calling the subroutine. The steel hysteretic model can be seen in Figure 7.

Set $\beta =(\varepsilon_{m}-\varepsilon_{0})/\varepsilon_{y}$ [45]. The unloading stiffness calculation formula is as follows:(19)$$E_{sr}=\left\{\begin{array}{c} E_{s}\\ (1.05-0.05\beta )E_{s}\\ 0.85E_{s} \end{array}\right.\begin{array}{c} ,\beta <1\\ ,1\leq \beta \leq 4\\ ,\beta >4 \end{array}$$

The model loading path formula is shown as follows:(20)$$\sigma =\gamma ({\bar{\varepsilon }}^{3}-{\bar{\varepsilon }}^{2})+(1-\alpha )\sigma_{m}\bar{\varepsilon }+\alpha \sigma_{m}$$
(21)$$\gamma =E_{sh}(\varepsilon_{m}-\varepsilon_{L})-(1-\alpha )\sigma_{m}$$
(22)$$\bar{\varepsilon }=(\varepsilon -\varepsilon_{L})/(\varepsilon_{m}-\varepsilon_{L})$$
where αis the influence coefficient of structural energy dissipation capacity, and the mathematical expression is as follows:(23)$$\alpha =\left\{\begin{array}{c} 0.5\\ (20-\beta )/38\\ 0 \end{array}\right.\begin{array}{c} ,\beta \leq 1\\ ,1<\beta <20\\ ,\beta \geq 20 \end{array}$$

### 3.2. Interface Simulation

Contact between different concrete surfaces was simulated by the Coulomb friction model. The tangential friction function was a penalty friction function with a friction coefficient of 0.6, and the rest of the behavior was defined as a hard contact. In order to save the settlement costs, the contact surface between the prefabricated column and the post-pouring core area, the contact surface between the inner surface of the keyway and the post-pouring concrete, etc., which had less influences on the predicted results, were set as the tie-type connections.

The embedded element method was used to simulate the interaction between the steel and concrete materials, but the bond slip between them was ignored. In order to reduce the simulation error, the reinforcement subroutine was used. 

In order to make the simulation closer to the actual situation, the surface of the angle steel and the side of the precast concrete column, the bottom surface of the precast concrete beam, the bolt hole of the angle steel and the bolts surface, and the upper surface of the bolts and the surface of the L-shaped angle steel were set as surface-to-surface contacts, the bolts embedded in the concrete were set to be embedded, the nuts and the bolts were connected by tie-connections.

### 3.3. Boundary Condition

In this experiment, the bottom of the specimen was hinged, and the beam end was constrained by the support bar to move up and down. Therefore, in the finite element model, the top surface and the bottom surface of the precast concrete column and the midpoint of the surface at outer end of the beam were taken as the reference points, and the reference points and the corresponding surfaces were constrained together by the coupling function. By limiting the degree of freedom of the selected surface, the boundary conditions were simulated, as shown in Table 2.

### 3.4. Unit Selection and Meshing

The concrete, L-shaped steel and the high-strength bolt involved in this paper were simulated using the solid elements of C3D8R, the steel bars was simulated by adopting the truss elements of T3D2.

The rationality of element meshing is related to the accuracy and convergence of the calculation results. Therefore, for three-dimensional solid elements, they should be divided into regular hexahedrons. The model in this paper adopted a mesh size of 100 mm for the division of concrete elements, 50 mm for the division of steel elements, 15 mm for the division of L-shaped steel, and 10 mm for the division of high-strength bolts. The specific divisions can be seen in Figure 8.

### 3.5. ABAQUS Parameter Setting

The introduced steel bar subroutine needed the input of elastic modulus, yield strength and the ratio of post-yield elastic modulus to the initial elastic modulus of rebar, detailed data can be seen in Table 3.

The plastic damage constitutive model of concrete adopted Linbliner yield surface and non-associated flow rule. Therefore, a series of plastic parameters should be defined, as shown in Table 4.

### 3.6. Analysis Results of the Finite Element Model

To prove the rationality of the finite element model proposed in this paper, the hysteresis curves and failure patterns obtained via finite element analysis were compared with the experimental results.

As shown in Figure 9 and Table 5, it can be seen that the hysteresis curves obtained via the simulation analysis are highly coincident with the experimental results. In the test, the bond slip phenomenon was more obvious, so the experimental curve was more pinched. The relative error between the final displacement and the ultimate displacement is less than 5%. Therefore, the finite element model established in this paper can accurately reflect the mechanical properties of the joints under the actual stress conditions.

Furthermore, the model proposed by Vaiana and Rosati [46] is important for the nonlinear dynamic analysis of real tridimensional structures. Therefore, this paper verifies the model by fitting the upper and lower limit curves obtained from simulation, it is found that the results of this paper are basically consistent with its model theory. The schematic diagram can be seen in Figure 10.

The fitting formula is as follows:(24)$$C_{u}=-6.436e^{0.02454x}+27.7+\frac{114.48}{1+e^{-2.595\left(x-6.155\right)}}+111.52+1.444x+28.28$$
(25)$$C_{u}=1115e^{-0.004321x}-229.5+\frac{72.28}{1+e^{-0.9502\left(x-0.4387\right)}}+918.2+6.879x-229$$

From the crack propagation mode shown in Figure 11, it can be found that the damage mode of the specimen was beam end bending damage. Cracks were mainly found in the joints between the new and the old concrete parts. The RPC was poured into the panel region and the keyway without causing any damages. The damage mainly occurred near the L-shaped angle steel, and extended to the outer end of precast beam. The surface of the beam displayed visible cracks, but there was no significant spalling of the concrete. Through the finite element analysis, it can be found that the joint failure mainly occurred near the shear key, and the column was not damaged. The comparison revealed that the finite element analysis can accurately reflect the mechanical properties of the prefabricated components. 

For structures with good seismic performance, yielding of the longitudinal reinforcement tends to occur first in the section of the maximum bending moment, and the so-called plastic hinge appears [47]. The presence of plastic hinge at the beam-to-column connections can reduce the ability of the frame to resist damage. Therefore, moving the “plastic hinge” outward can enhance the joint’s ability to resist destruction.

The maximum stress of the steel bars of the joint appeared on the outside of the L-shaped steel, as shown in Figure 12. It illustrates that the deformation resistance of the structure is enhanced by adding the L-shaped steel, and the design purpose of shifting the plastic hinge of the beam end and improving the seismic performance of the joint is achieved.

## 4. Parametric Analysis

It can be seen from the previous verification that the finite element model can accurately reflect the seismic behavior of beam–column joints of assembled local high-performance concrete frame. Therefore, without changing the framework of the finite element model, the mechanical properties of the prefabricated joints under different conditions can be obtained by changing the relevant parameters. The main parameters are the axial compression ratio of the column top, the spacing of the stirrups in the core area, the diameter of the PC steel bar, and the strength of the RPC.

### 4.1. The Column Top Axial Pressure Ratio

It can be concluded from Figure 13 that the load–displacement curves of the joints remain consistent under different axial compression ratios, and the ultimate loads are relatively close. As the test continues, the bearing capacity of the models with different axial pressure ratios tends to be almost equal to each other. Therefore, it has minimum effect on the mechanical properties of the structures.

### 4.2. Stirrup Spacing in Core Area

By observing Figure 14, it can be found that as the stirrup spacing in the panel area decreases, the ductility of the structure increases and the ultimate bearing capacity also increases. By comparing the members with stirrup spacing of 100 mm and 150 mm, it can be concluded that casting RPC into the core region and keyway can improve the ability of the joints to resist shear mode failure while reducing the number of stirrups. When the stirrups are arranged at a 200 mm spacing, the bearing capacity of the joints decreases more significantly.

### 4.3. Diameter of PC Steel Bar

From Figure 15, it can be found that with the increase in the diameter of the PC steel bars, the ultimate bearing capacity of the joints also increases. The load of the PC steel bars with a diameter of 16 mm and 20 mm decreases slowly, and the ability to dissipate energy is better. The load of the component with a diameter of 12.6 mm decreases more obviously.

### 4.4. The RPC Strength Grade

To compare the influence of the concrete strength poured in the core area and keyway on the performance of the joint, on the basis of the RPC strength grade being changed to 130 MPa and 160 MPa, a group of C40 ordinary concrete is added as the comparison group.

From Figure 16, it can be concluded that the bearing capacity of the structures is positively correlated with the RPC strength grade, and it is much higher than that of the component pouring C40 concrete. When the RPC strength is 100 MPa and 130 MPa, the descending section of the skeleton curve tends to be gentle. When the RPC strength grade reaches 160 MPa, the joint decreases most significantly after reaching the ultimate bearing capacity, indicating that the ductility of the joint decreases. Therefore, pouring a certain strength of RPC in the keyway and the core area can improve the ductility and energy dissipation capacity of the joints, thereby improving the seismic behavior of the joints.

## 5. Conclusions

The corresponding finite element model was established according to the test components, and the finite element calculation results were compared with the test. The finite element model can accurately simulate the mechanical performance and failure mechanism of the structures. After proving the accuracy of the model, some parameter analyses were carried out. The following conclusions can be drawn from the numerical results obtained via experiments and simulations.

(1) Installation of the L-shaped angle steel can effectively improve the capacity to dissipate energy and the integrity of the precast joints. The plastic hinge of the beam end can be shifted outward to avoid premature destruction of the core area.

(2) The failure mode of the component is the bending failure at the beam end, and there is no obvious damage to the column. It shows that the keyway and the core area pouring RPC can improve the peak load and the failure resistance of the joints.

(3) The stirrup spacing can be appropriately increased in the area of pouring RPC. However, when the stirrup spacing is too large, the bearing capacity of the joints will be greatly reduced. 

(4) The ultimate bearing capacity of precast joints increases with the improvement of PRC strength in the post-cast area. After the specimen yields, the bearing capacity decreases faster with higher RPC strength in the post-cast area of the joints.

(5) The axial compression ratio of the column has a minimum effect on the mechanical performance of the joints. As the force-bearing steel bars at the bottom of the precast concrete beam, increasing the diameter of the PC steel bars can effectively improve the bearing capacity of the joints. Therefore, increasing the diameter of the PC steel bars appropriately can improve the mechanical performance of the joints.

(6) The determination of the limit state of reinforced concrete cross-section is of great significance to ensure the safety of the structures. Therefore, it is necessary to use the probability evaluation method [48] to determine the limit state in the future work.

## Figures and Tables

**Figure 1 materials-17-00107-f001:**
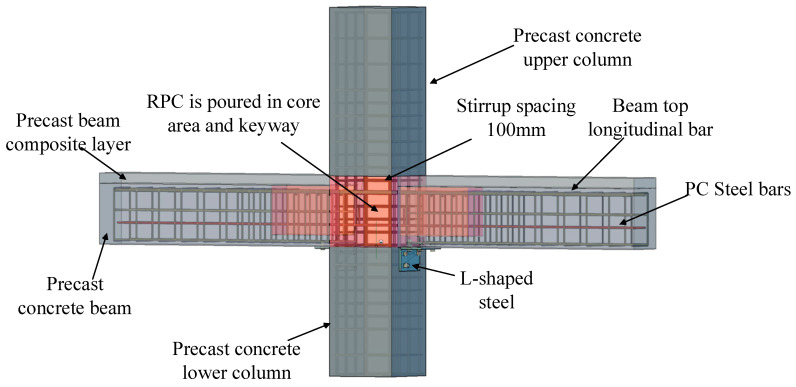
Precast beam–column joint.

**Figure 2 materials-17-00107-f002:**
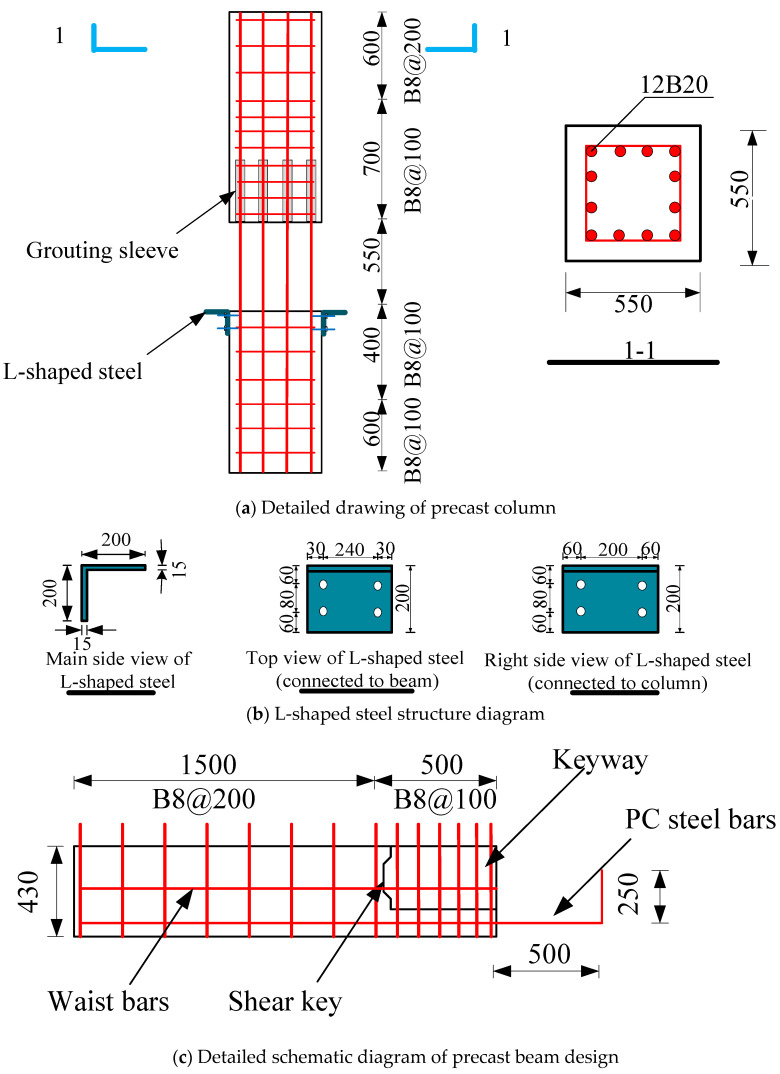
Detailed diagram of joint design. Note: B = HRB400 steel bars; D = PC steel bars.

**Figure 3 materials-17-00107-f003:**
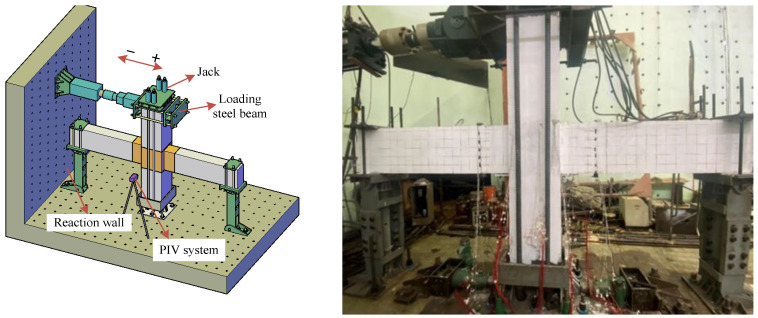
Loading device diagram.

**Figure 4 materials-17-00107-f004:**
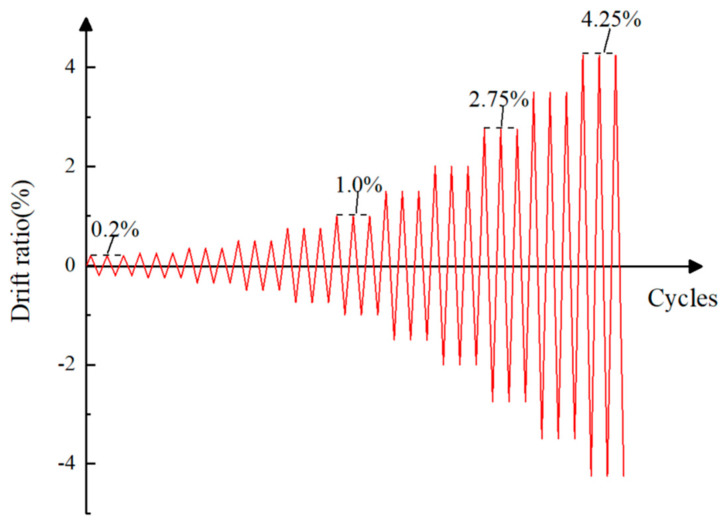
Loading system diagram.

**Figure 5 materials-17-00107-f005:**
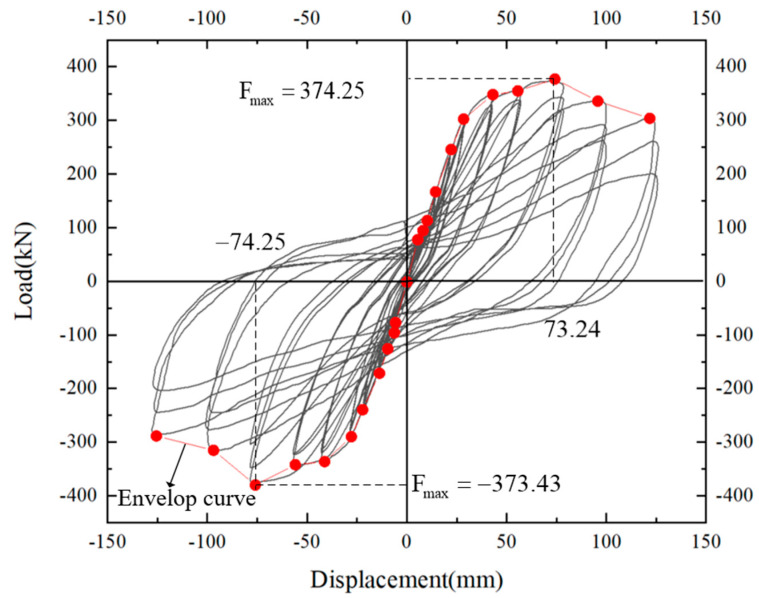
Hysteresis curve and envelop curve of the joint.

**Figure 6 materials-17-00107-f006:**
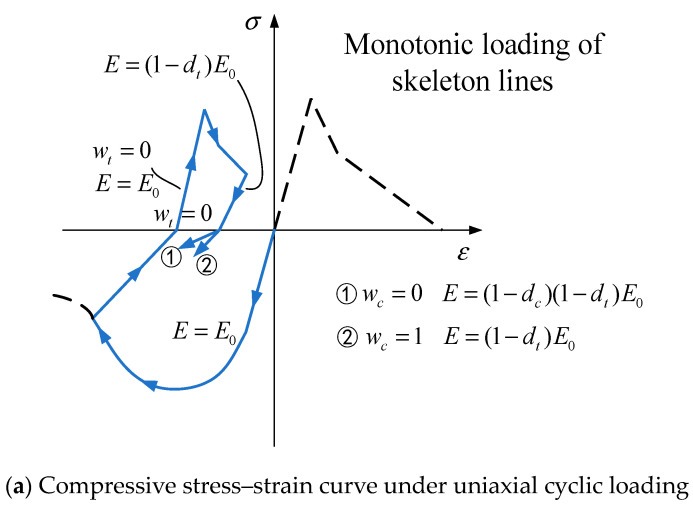
Stress–strain curve of the concrete plastic damage model under uniaxial cyclic loading. Note: Blue curve represents compression; red curve represents tension.

**Figure 7 materials-17-00107-f007:**
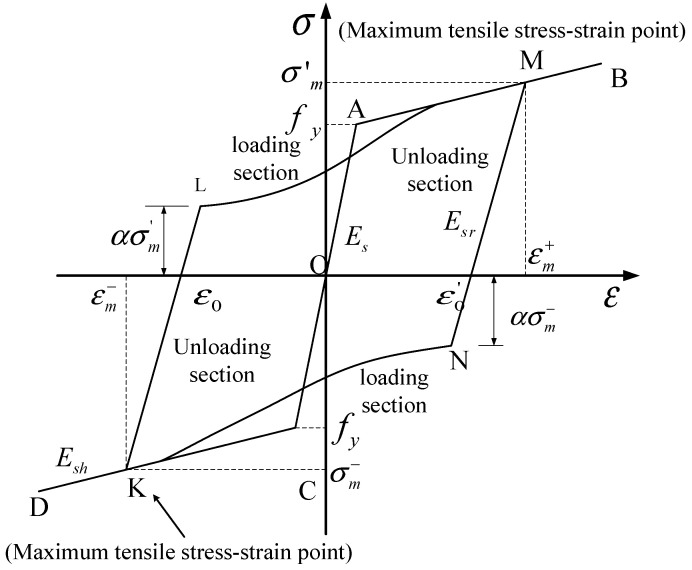
The steel hysteretic model.

**Figure 8 materials-17-00107-f008:**
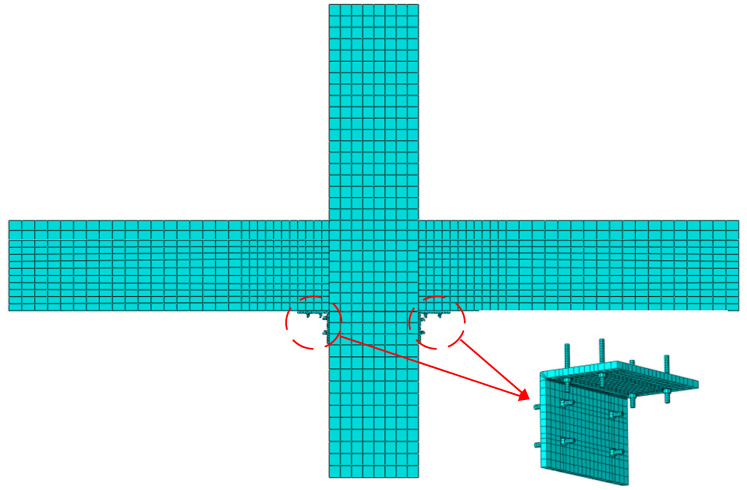
The finite element model of beam–column joints of assembled local high-performance concrete frame.

**Figure 9 materials-17-00107-f009:**
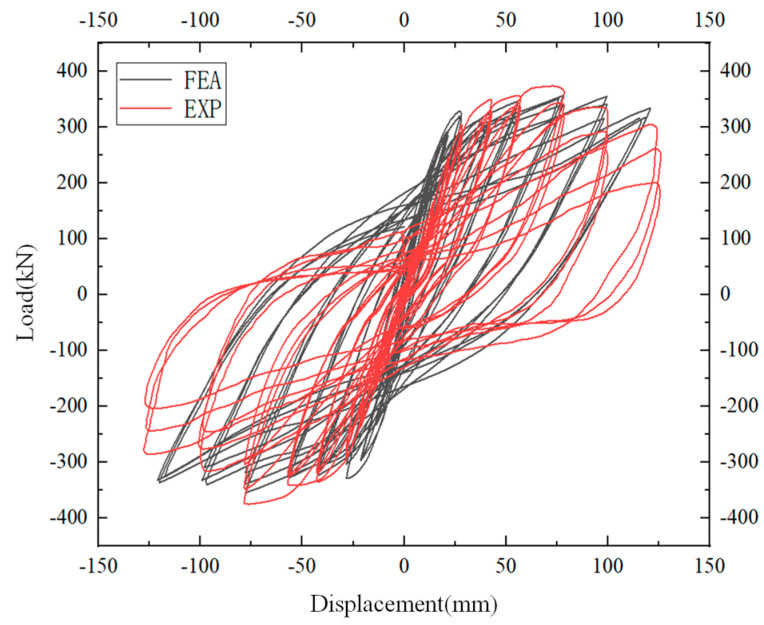
Hysteresis curve comparison diagram.

**Figure 10 materials-17-00107-f010:**
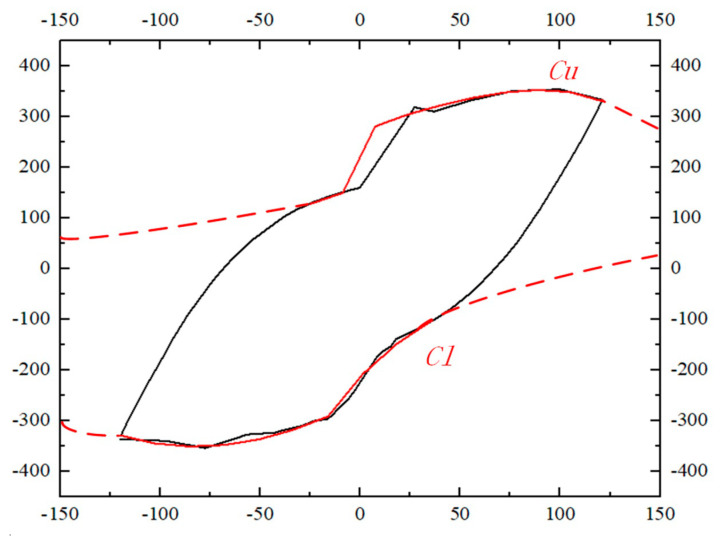
The upper and lower limit curve diagram. Note: *Cu* = upper limit curve; *Cl =* lower limit curve.

**Figure 11 materials-17-00107-f011:**
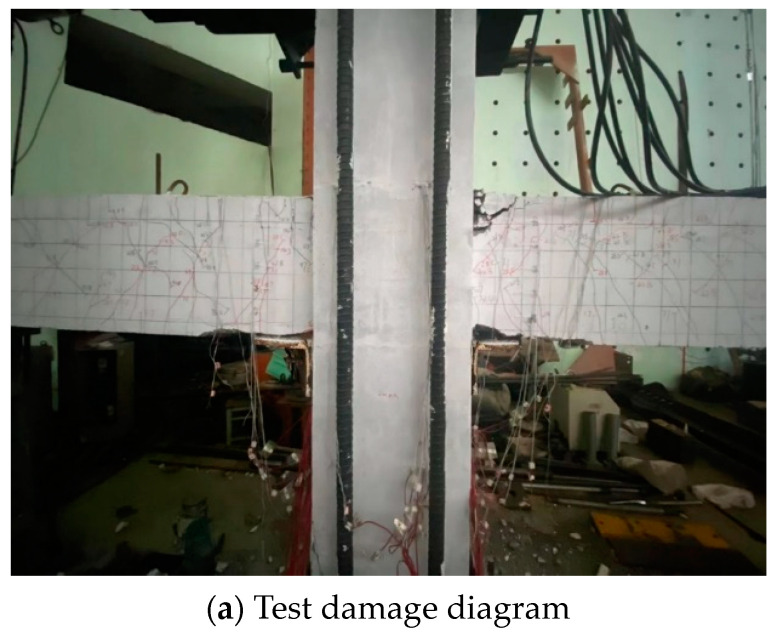
Specimen damage diagram.

**Figure 12 materials-17-00107-f012:**
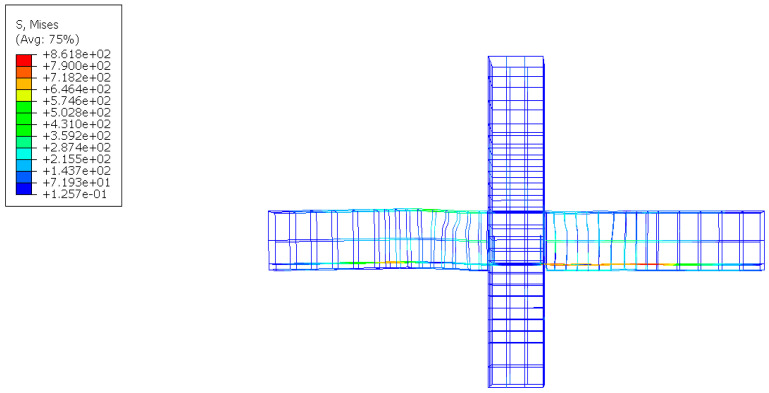
Steel stress cloud diagram.

**Figure 13 materials-17-00107-f013:**
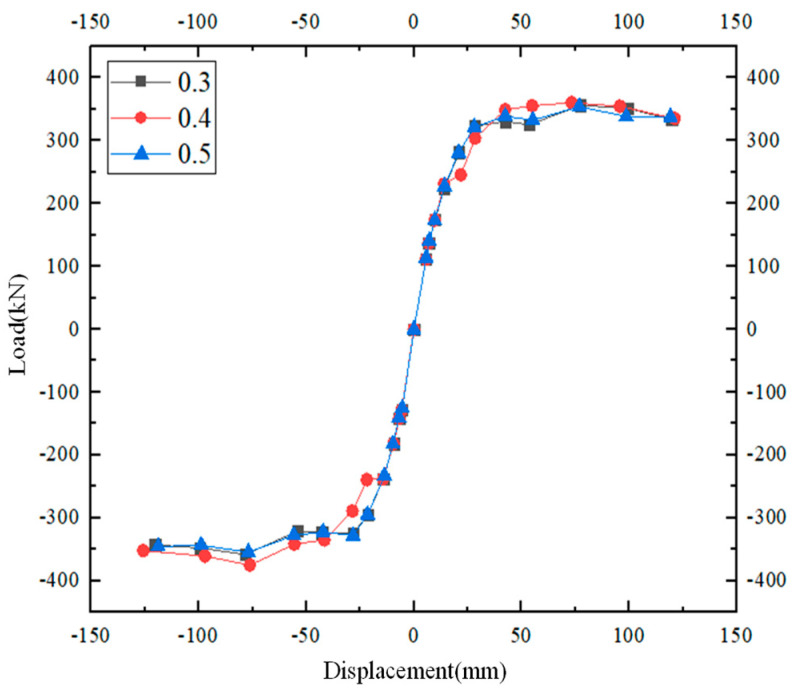
Load–displacement curves under different axial compression ratios.

**Figure 14 materials-17-00107-f014:**
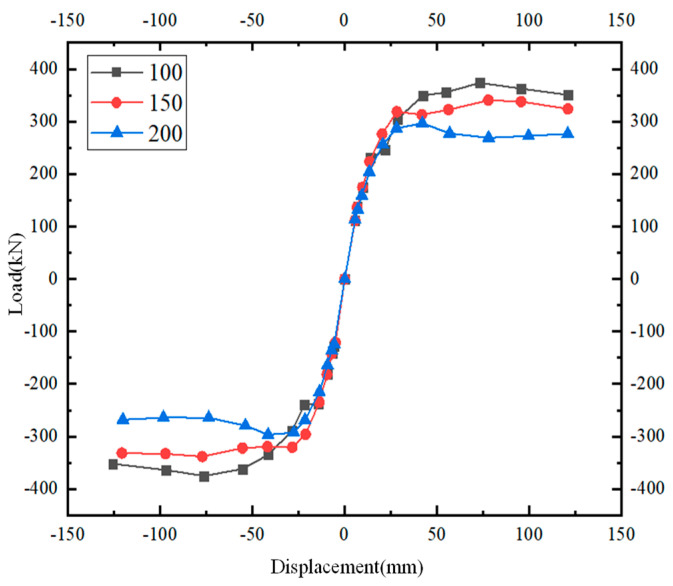
Load–displacement curves of specimens under different stirrup spacing.

**Figure 15 materials-17-00107-f015:**
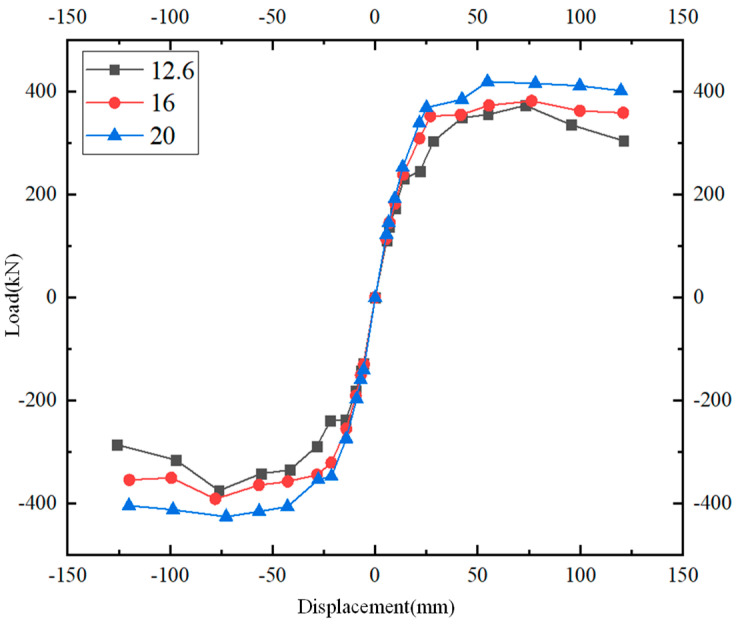
Load–displacement curves under different PC steel bar diameters.

**Figure 16 materials-17-00107-f016:**
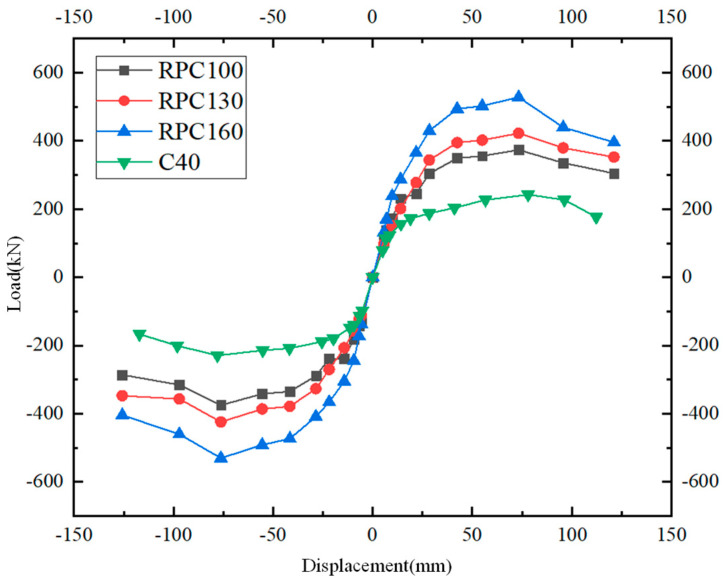
Load–displacement curves under different concrete strength grades.

**Table 1 materials-17-00107-t001:** Horizontal loading displacement control value.

Loading Level	Drift Ratio (%)	Displacement (mm)	Loading Stage
		2	Preload
1	0.2	5.7	Formal loading
2	0.25	7.1	
3	0.35	9.9	
4	0.5	14.3	
5	0.75	21.4	
6	1	28.5	
7	1.5	42.8	
8	2	57.0	
9	2.75	78.4	
10	3.5	99.8	
11	4.25	121.1	

**Table 2 materials-17-00107-t002:** Boundary condition.

Bearing Position	Degree of Freedom					
	U1	U2	U3	UR1	UR2	UR3
Top of column	L	R	R	R	R	L
Bottom of column	R	R	R	R	R	L
Outer end of beam	L	R	R	R	R	L

Note: ‘R’ denotes the constraint, and ‘L’ denotes the unconstraint.

**Table 3 materials-17-00107-t003:** ABAQUS steel constitutive model parameters.

Steel Grade	Diameter/mm	Elastic Modulus $E_{s}$/MPa	Yield Strength $f_{y}$/MPa	Elastic Modulus Ratio
HRB400	8	2.0 × 10^5^	435	0.001
HRB400	10	2.0 × 10^5^	441	0.001
HRB400	20	2.0 × 10^5^	449	0.001
PC bar	12.6	1.95 × 10^5^	1466	0.001

**Table 4 materials-17-00107-t004:** Related parameters of the concrete damage plasticity model.

Parameter	Dilation Angel	Eccentricity	$f_{bo}/f_{co}$	$k_{c}$	Viscous Coefficient
Numerical Value	38	0.1	1.16	0.6667	0.0005

**Table 5 materials-17-00107-t005:** Comparison of simulation and test termination displacement and ultimate load.

Comparison Type	Load Direction	Experiment	Finite Element	Relative Error (%)
Ultimate displacement (mm)	+	124.24	121.02	−2.59
	-	125.72	120.19	−4.40
Ultimate load (kN)	+	373.63	355.26	−4.92
	-	373.43	354.84	−4.98

## Data Availability

Data available on request due to restrictions e.g., privacy of ethical.

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
