# Peer review of "Research on Nonlinear Behavior of Local High-Performance Concrete Beam–Column Connections"

_materials, 2023, doi:10.3390/ma17010107_

Round 1

Reviewer 1 Report

Comments and Suggestions for Authors

The paper entitled “Research on seismic behavior of local high-performance concrete beam-column connections” proposes a topic of current interest that is suitable for the Materials journal. However, there are several issues to be addressed before considering it for publication:

1) The title of the paper should be properly reformulated. A possible suggestion is: “On the nonlinear behavior of local high-performance concrete beam-column connections”.

2) The abstract is quite long so it is highly suggested to shorten it.

3) In the keywords, the “Hysteretic model of steel bar” is too long and can be divided into two separate words.

4) At page 6, there is a red word, that is, integral. Why?

5) At page 7, the following sentence should be revised: “ finally s an inverted S-shape is shaped ”.

6)     In Section 3.6, the authors compare the experimental and simulated hysteresis loops which refer to the input displacement illustrated in Figure 4. What does it happen if one applies a random displacement as it happens during earthquake excitations? Does the proposed model still work?

7) At the end of Section 3 it is crucial to explain that the proposed finite element model results can be also used to calibrate accurate phenomenological models available in the literature [A] since the latter are extremely useful to perform nonlinear dynamic analyses of real tridimensional structures.

[A] https://doi.org/10.1016/j.ymssp.2023.110448

8)     In the conclusion section, the authors should refer to the need of performing, in future work, a probabilistic assessment by using sophisticated methods already available in the literature [B].

[B] https://doi.org/10.1007/s11012-019-00979-4

Comments on the Quality of English Language

none

Reviewer 2 Report

Comments and Suggestions for Authors

The purpose of this study is to enhance the understanding of various parameters for improving the seismic performance index of precast structures. Using the ABAQUS platform, the authors established a finite element model based on the experimental configuration to analyze the seismic performance index of the precast structure. After confirming a high degree of agreement between the finite element calculation results and the experimental results, finite element analysis supplemented aspects that are challenging to address through experimental analysis alone. To enhance the joint's performance, the authors aimed at improving the understanding of parameters such as the axial compression ratio of the precast column, the core stirrup spacing, the diameter of the PC steel bar, and the concrete strength of the core and keyway areas. The findings of this study offer both experimental evidence and theoretical foundations, laying the groundwork for future research aimed at enhancing the performance of precast concrete joints. However, researchers should make some adjustments to improve grammatical errors and readability, taking into account the comments below.

·         On page#2, line#33, There are no full spellings of ‘RPC’. the author should provide the full spelling of the abbreviations when they appear first in the manuscript.

·         On page#3, line#9, ‘[email protected] cross section’ requires a space after ‘200.’.

·         On page#3, line#11, The verb “is” should be changed to “was”. This is because the paper used the past tense in the previous sentence.

·         On page#4, The images of Figure.1, it is difficult to grasp the meaning of the under bar under the sentence and 1-1 under the column section.

·         On page #5, line #5, Should change 'ration' to 'ratio', which appears to be a typo.

·         On page #6, line #9, Can't find the reason why 'integral' is highlighted in red.

·         On page #8, The formulas don't look evenly arranged.  

·         On page#15, The image in Figure.11, the resolution of the image is low, so it looks blurry.

·         On page#19, The arrangement and size of letters and sentences are uneven in the annotation section.

Comments on the Quality of English Language

The English used in the research paper is generally well written, and the content is clearly communicated.

Reviewer 3 Report

Comments and Suggestions for Authors

Reviewer's report on the paper titled “Research on seismic behavior of local high-performance concrete beam-column connections”

The paper “Research on seismic behavior of local high-performance concrete beam-column connections” is an original contribution, which corresponds to the scope of MDPI Journal Materials. A detailed parameter analysis of prefabricated reinforced concrete frame joints is carried out by setting different axial pressure ratio of precast column top basing on the validated model developed on the ABAQUS platform. The analysis results illustrate that the different axial pressure ratio at the top of the column has minimal impact on the joint's performance. Results of this analysis can provide experimental basis and theoretical support for the further investigation of the node in the later stage . The paper was written at the good technical level and enough clear for understanding. But the following comments should be addressed at the same moment:

1. Chapter 1. Introduction should be expanded by the clear definition of goal of the current investigation and its basis. Clear formulations of the tasks should be solved to obtain the above mentioned goal should be clearly formulated also. 

2. Uniaxial tensile constitutive of concrete, described in the sub-chapter 3.1.1. by the equations 1) – 9), should be supplied by the some textual explanations also. 

3. Figure 6 should be supplied by the designations of colors below the chart.

Diameter and length of the fasteners for the L-shaped steel meber on the Fig.8 should be mentioned in the text or in the designations below the figure. 

4. Chapter 5. Conclusions should be supplied by the numerical results, obtained in the course of the current investigation and basing the major generalizations are done.

Reviewer 4 Report

Comments and Suggestions for Authors

In this article "Research on seismic behavior of local high-performance con crete beam-column connections", written by authors Xu Zhiqiang, Jianbing Yu, Yufeng Xia and Chaojun Jiang, the issues of "cold joints" in connecting prefabricated linear reinforced concrete elements with monolithic reinforced concrete frames were addressed.

The authors focused their research on the joint and many possible connection options.

Experimental tests and computer simulations were performed using the very popular ABAQUS program, which was accompanied by additional programs enabling a better numerical description of the interaction between concrete and reinforcing steel.

Constitutive models for concrete were used in accordance with global standards, including ACI.

The design of the load-bearing capacity of the joint connecting the prefabricated elements was based on the principle of 'strong joints and weak members, strong shear and weak bending '.

Of course, there are also other solutions, e.g. the use of very flexible reinforcing bars in horizontal beams near the beam-column connection.

During the experiment and in the simulation in the numerical program, different types of concrete were used, from ordinary to RPC, and different types of reinforcing steel diameters and different stirrup spacing in the node.

Variable cyclic loads simulating seismic loads were programmed in the experiment and in the ABAQUS program.

The conclusions drawn after the experiment and numerical calculations are intuitively correct and, of course, interesting.

Issues related to "cold joints" were, for example, recognized in connection with the pedestrian bridge catastrophe  on the university campus in Miami.

Additional problems that arise when using "cold joints" are:

  - difference in humidity of prefabricated elements and concrete mortar - drying of the mortar,

  - preparation of the surface in the joint, the surface of the prefabricated element and the monolithic frame, e.g. removal of cement powder,

  - use of intermediate coatings - primers,

  - vibration of the concrete mortar in the joint,

  - the influence of fibers on the connection between concrete and steel.

These are, of course, known physical phenomena, but poor awareness of the influence of the mentioned parameters on the load-bearing capacity of the joint was one of the important causes of the disaster in Miami, where the clod joint was sheared.

The article is interesting and the conclusions are very  clearly recommended  spreading them.

Round 2

Reviewer 1 Report

Comments and Suggestions for Authors

The authors addressed all the issues identified by this reviewer. I suggest the Editor to accept the manuscript only after the authors will address the following issue:

In the reference section, it is necessary to correct references 47 and 49 since they are currently incorrect. The correct version is:

Ref. 47:

Vaiana, N., Rosati, L. (2023). Analytical and differential reformulations of the Vaiana–Rosati model for complex rate-independent mechanical hysteresis phenomena. Mechanical Systems and Signal Processing199, 110448.

Ref. 49:

Sessa, S., Marmo, F., Vaiana, N., Rosati, L. (2019). Probabilistic assessment of axial force–biaxial bending capacity domains of reinforced concrete sections. Meccanica54, 1451-1469.

Comments on the Quality of English Language

none